# Measurement Report: Lidar measurements of stratospheric aerosol following the 2019 Raikoke and Ulawun volcanic eruptions

Geraint Vaughan[1], David Wareing[2], and Hugo Ricketts[1]

[1]National Centre for Atmospheric Science, University of Manchester, UK
[2]Aberystwyth University, UK

**Correspondence:** Geraint Vaughan (geraint.vaughan@ncas.ac.uk)

**Abstract.** At 18 UTC on 21 June 2019 the Raikoke volcano in the Kuril islands began a large magnitude explosive eruption, sending a plume of ash and sulphur dioxide into the stratosphere. A Raman lidar system at Capel Dewi Atmospheric Observatory, UK, was deployed to measure the vertical extent and optical depth of the volcanic aerosol cloud following the eruption. The elastic channel at 355 nm allowed measurements up to 25 km, but the Raman channel was only sensitive to the tropo-
sphere. Therefore, retrievals of backscatter ratio profiles from the raw backscatter measurements required aerosol-free profiles derived from nearby radiosondes and allowance for aerosol extinction using a lidar ratio of 40-50 sr. Small amounts of aerosol were measured prior to the arrival of the volcanic cloud (27 June – 5 July 2019), from pyroconvection over Canada. Model simulations by de Leeuw et al. (2020) and Kloss et al. (2020) show that volcanic ash may have reached Europe from 1 July onwards, and was certainly present over the UK after 10 July. The lidar detected a thin layer at an altitude of 14 km late on 3
July, with the first detection of the main aerosol cloud on 13 July. In this initial period the aerosol was confined below 16 km but eventually the cloud extended to 20.5 km. A sustained period of clearly enhanced stratospheric Aerosol Optical Depths began in early August, with maximum value (at 355 nm) around 0.05 in mid-August and remaining above 0.02 until early November. Thereafter, optical depths decayed to around 0.01 by the end of 2019 and remained around that level until May 2020. The altitude of peak backscatter varied considerably (between 14 and 18 km) but was generally around 15 km. However, on one
notable occasion on 25 August 2019, a layer around 300 m thick with peak lidar backscatter ratio around 1.5 was observed as high as 21 km.

## 1   Introduction

From 1805 UTC on 21 June 2019 to 0540 UTC on 22 June the Raikoke volcano in the Kuril Islands (48.29°N, 153.25°E)
erupted, sending plumes of ash and sulphur dioxide into the stratosphere (Crafford and Venzke, 2019). With an estimated 1.5 $\pm$ 0.2 Tg of $SO_2$ (de Leeuw et al., 2020), it was one of the largest injections of volcanic aerosol into the stratosphere since the Pinatubo eruption in 1991 and created vivid sunsets around the northern hemisphere (Fox, A., 2019). Sulphur dioxide was

measured from 11 to 20 km by the TROPOMI instrument on the SENTINEL-5 satellite on 24 June, with ash detected by the CALIOP spaceborne lidar on the CALIPSO satellite at 17 km on 22 June and around 13 km on 23 and 24 June (Hedelt

et al., 2019). The ash however was quickly dispersed or sedimented from the stratosphere in the week following the eruption, leaving the cloud of $SO_2$ to oxidise to sulphuric acid and spread around the hemisphere in aerosol form. Maximum average stratospheric aerosol optical depths at 449 nm (derived from SAGE-III/ISS data) reached 0.045 north of 55°N and 0.030 between 40 and 55°N in the early months after the eruption (Kloss et al., 2020).

Lidar measurements from Hawaii measured a layer of aerosol around 1 km thick at 26 km on 24 September 2019 (Chouza

et al., 2020); these authors also found using CALIOP data that the layer had ascended from around 19 km in the previous 2 months. During the months following the initial eruption the aerosol evolved in both height, depth and optical thickness (Kloss et al., 2020). It also merged with aerosol from the smaller eruptions of Ulawun in Papua New Guinea (5.05°S, 151.3°E) that occurred on 26 June and 3-4 August 2019, which reached 19 km altitude and injected around 0.14 and 0.2 Tg $SO_2$ respectively into the stratosphere (Kloss et al., 2020).

Lidar measurements of the volcanic aerosol cloud at 355 and 532 nm from four Russian stations were presented for the second half of 2019 by Grebennikov et al. (2020). These stations ranged in longitude from Obninsk at 36.6°E to Petropavlovsk-Kamchatsky at 158.65°E, and observed volcanic aerosol from late July onwards, reaching up to 18-20 km. A maximum integrated backscatter at 532 nm above 13 km of $> 10^{-3}$ was found in August, corresponding to aerosol optical depth of around 0.045.

Measurements are presented here from a Raman lidar system based at the Capel Dewi Atmospheric Observatory, UK (52.4°N, 4.1°W), beginning from June 2019 and continuing until the Spring of 2020, showing how the aerosol cloud evolved over the lidar site. All the measurements were taken during the hours of darkness when there was no cloud cover over the site; in all there were 34 nights' measurements between 27 June 2019 and 30 May 2020.

## 2   Method

The Capel Dewi Raman lidar system (Vaughan et al., 2018) operates in the ultraviolet at 355 nm using a Continuum Powerlite 9030 laser with a pulse energy of 300 mJ and a pulse repetition frequency of 30 Hz. The receiver system is usually optimised for measuring signals above 2 km and has three interference filters to measure the elastic backscatter (355 nm), nitrogen Raman scattering (387 nm) and water vapour scattering (408 nm). Photon-counting electronics are used with range gates of 100 ns, providing a range resolution of 15 m. To enhance sensitivity in the elastic channel, a neutral density filter was removed, which

extended the measurement range to around 25 km in the lower stratosphere, but raising the lower limit to around 7 km.

Raw data was collected with a time resolution of 10 minutes on most nights, and the files combined to whole-night averages for further analysis. Filters were applied during averaging to remove files affected by low cloud to guard against signal-induced noise problems. Analysis proceeded by converting the elastic signal profiles to lidar backscatter ratios - the ratio of the total backscatter profile to that which would be returned by a pure molecular atmosphere. The optimum way to accomplish this is to

use data from the $N_2$ Raman channel, as in Vaughan et al. (2018), as this automatically allows for attenuation of the signals by

the aerosol layer. However, the faint signals on the $N_2$ Raman channel in the lower stratosphere meant that long runs of data had to be combined to accumulate enough signal for analysis. This was only possible for a few nights during the period under consideration here.

Therefore, for each night of measurement, a density profile from a nearby radiosonde ascent (chosen using the wind direction at 200 mb) was used to construct a molecular backscatter profile, which was fitted to the elastic signal above the aerosol layer on that night (usually above ∼20 km). An onion-peeling retrieval with prescribed lidar ratio (ratio of aerosol extinction to backscatter coefficient) was then used to derive the lidar backscatter ratio down to the upper troposphere, as used in Thomas et al. (1987) and Vaughan et al. (1994). This algorithm sequentially removed the attenuation due to the aerosol, layer by layer, beginning from the aerosol-free fitting region and ending at 6 km altitude. As cirrus clouds were frequently observed near the tropopause, the algorithm used two layers with different lidar ratio: a stratospheric value above 12 km (or above the cirrus layer if this was higher) and a different, usually lower value, below this height. (During the period up to 6 July, absorbing aerosol was found near the tropopause, necessitating a larger value of lidar ratio, see section 4).

For volcanic aerosol clouds from very large magnitude eruptions (e.g. Pinatubo in 1991) the lidar ratio is variable, for example due to variations in the size distribution of the sulphuric acid aerosol (e.g. Vaughan et al. (1994)). For moderate magnitude eruptions, Mattis et al. (2010), using Raman lidar measurements for small volcanic plumes in 2008-9, found a range of values from 30 to 60 sr at 355 nm. Ash tends to increase the lidar ratio: Lopes et al. (2019) quoted $63 \pm 21$ sr for the Calbuco eruption plume of 2015; Chouza et al. (2020) quoted $64 \pm 27$ sr for the Nabro plume of 2011 and Hoffmann et al. (2010) quoted $63 \pm 10$ sr for the Kasatochi plume of 2008. Remarkably consistent though these results are, Mie scattering calculations suggest that the lidar ratio depends strongly on particle size as well as composition (Korshunov and Zubachev, 2013) and therefore varies from eruption to eruption.

For nights when the volcanic aerosol plume formed a distinct layer, and there was no cirrus cloud in the troposphere, an appropriate value of lidar ratio could be found by requiring that the backscatter return to the molecular profile below the layer. For most of the period of this study this resulted in values around 40 sr, which has been adopted for most of the dataset for consistency. A cross-check on the lidar ratio was provided on a few nights where enough profiles could be combined to yield a useful signal on the Raman channel. This allowed an independent measure of the optical depth of the stratospheric aerosol layer, further discussed below.

One of the characteristics of the Capel Dewi lidar is that its receiver is only sensitive to signals whose polarisation is parallel to that of the incident laser beam. Thus, when non-spherical particles are present, the backscatter ratio is underestimated and the effective lidar ratio becomes artificially large. Nonetheless, the optical depth derived for these cases should be correct, as the two errors compensate for each other.

## 3  The Raikoke eruption

Following the eruption of the Raikoke volcano on 21-22 June 2019, the ash and sulphur dioxide plume initially moved eastward before being entrained in a cyclonic circulation over the North Pacific. The on-line NOAA HYSPLIT model (Stein et al., 2015)

provides a tool for estimating the dispersion of a volcanic plume and Figure 1 shows the simulated ash cloud 72 hours after the

90 eruption, confined to a region between Kamchatka and Alaska. Profiles from CALIOP (available from https://www-calipso. larc.nasa.gov/products/lidar/browse_images/std_v4_index.php, last access: 26 August 2020) confirm the location of the ash plume at this time, along an orbit between 56°N, 177.8°E and 61.5°N, 173.5°E, at altitude 12-14 km.

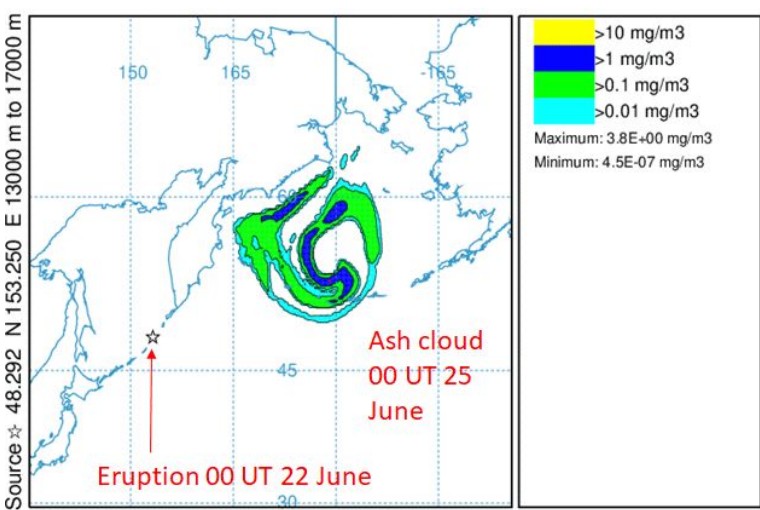

**Figure 1.** Dispersion of the volcanic ash cloud according to the Hysplit model 72 hours after the eruption. The model was initialised with a uniform ash injection between 13 and 17 km. Mass loadings are arbitrary and serve only to delineate the position of the cloud.

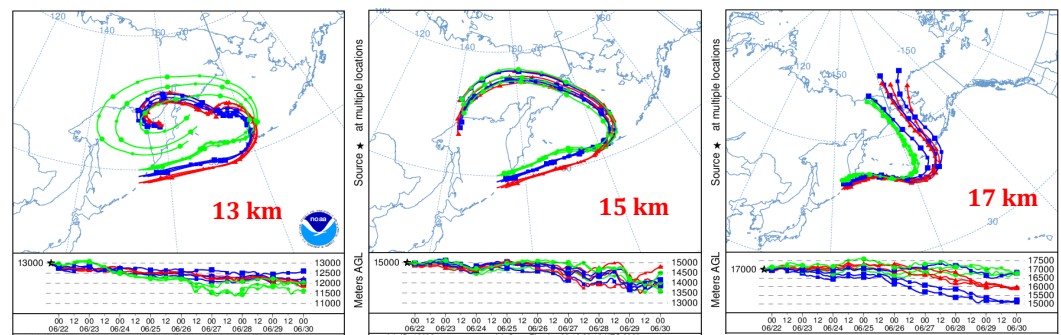

**Figure 2.** 8-day forward trajectories from nine points surrounding Raikoke, calculated using Hysplit and initialised 18 UT 21 June 2019, at three lower stratospheric heights

CALIOP profiles in the period 25-28 June show that the volcanic ash stayed broadly in the same region, becoming thin, patchy and generally confined below 15 km. The depolarisation ratio at 532 nm decreased from 30-40% on the 25th to below

20% by the 28th. HYSPLIT trajectory calculations (Figure 2) are consistent with the observations, suggesting little transport of material from the cyclonic circulation until the end of June; they are remarkably non-dispersive for 8-day trajectories. A further set of Hysplit trajectories (not shown), initialised from the end points of those in Figure 2, suggested the plume was confined until around 6 July, but these trajectories were more dispersive and cannot rule out a certain amount of transport westward in the first week of July.

The Hysplit model calculates air parcel trajectories based on 3-D advection by winds from an operational analysis model, and its predictions become increasingly sensitive to initial conditions as time goes on (e.g. Vaughan et al. (2018)). An alternative approach to simulating the spread of the aerosol cloud was presented by de Leeuw et al. (2020), using the UK Met Office's NAME dispersion model. This model is based on the global winds from the Met Office Unified Model analyses and includes chemical reactions for converting $SO_2$ to sulphate, as well as mixing through turbulence and subgridscale dynamics. Its simulations of $SO_2$ were found to agree well with the TROPOMI satellite for the three weeks after the eruption. de Leeuw et al. (2020) provide video files of model simulations as supplements to their paper, one of which shows the spread of volcanic aerosol across the Northern Hemisphere after the eruption. Up to the end of June the cloud was confined to North America and eastern Asia. Between 1 and 4 July there are hints that small amounts of aerosol were reaching Europe, with a more prominent filament reaching Scotland by the $7^{th}$. The main aerosol cloud in this simulation reached the southern UK on 10 July. These conclusions are consistent with the CLAMS model simulations presented by Kloss et al. (2020) (their fig.5), suggesting that lidar observations over Europe might detect volcanic aerosol from 1 July onwards, and would definitely do so after the $10^{th}$. The analysis of OMPS satellite data by Kloss et al. (2020) showed small amounts of stratospheric aerosol over Europe between 24 June and 6 July 2019 (their Fig. 3b), which they attribute using CLAMS modelling calculations to plumes from pyroconvection in Alberta.

We now turn to the lidar measurements at Capel Dewi and the arrival of the volcanic cloud in Europe.

## 4 Results

Lidar profiles at the end of June and the first few days of July 2019 showed numerous small aerosol layers in the lower stratosphere, consistent with the OMPS observations. An example, from the night of 1-2 July, is shown in Figure 3a. Two aerosol layers are shown in this figure - one around 12 km, just above the tropopause, which seems from its optical properties to have been smoke (a lidar ratio of 100 sr was needed to account for the attenuation of the laser beam through the layer) and another between 13 and 14 km where a lidar ratio of 40 sr sufficed. A CALIPSO orbit around 3° east of the lidar at 0250 UT on 2 July measured an aerosol layer around 12-13 km between 53.7°N and 55.5°N with depolarisation ratio 10-20%, confirming the presence of non-spherical particles in the lower layer, but showing no trace of the upper layer. The CALIPSO aerosol subtype algorithm (Kim et al., 2018) identified this as Type 10 (sulphate/other), which gives little clue to its identity.

A much more prominent aerosol layer was measured two days later (figure 3b); the maximum backscatter ratio was now 1.3 rather than 1.09 on 1 July. Again there are two layers, with the lower one more variable and more absorbing than the upper one. Both probably consisted of depolarising particles: CALIOP measured patches of depolarising aerosol ($\delta_v \approx 0.1 - 0.2$) at

52.0°N, 4.4°E, both at 11 and 14 km, at 0220 UT on 4 July (figure 4). The aerosol subtype algorithm gave Type 10 for the lower layer around 52°N. The aerosol in Fig.3 is therefore most likely due to pyroconvection, but it is not possible to rule out a contribution from volcanic aerosol over Europe at this time. To emphasise the patchy nature of the observations in early July, there was no aerosol above the tropopause on the following night (4-5 July) and only a faint layer on 5-6 July (12 - 14 km, maximum backscatter ratio 1.03).

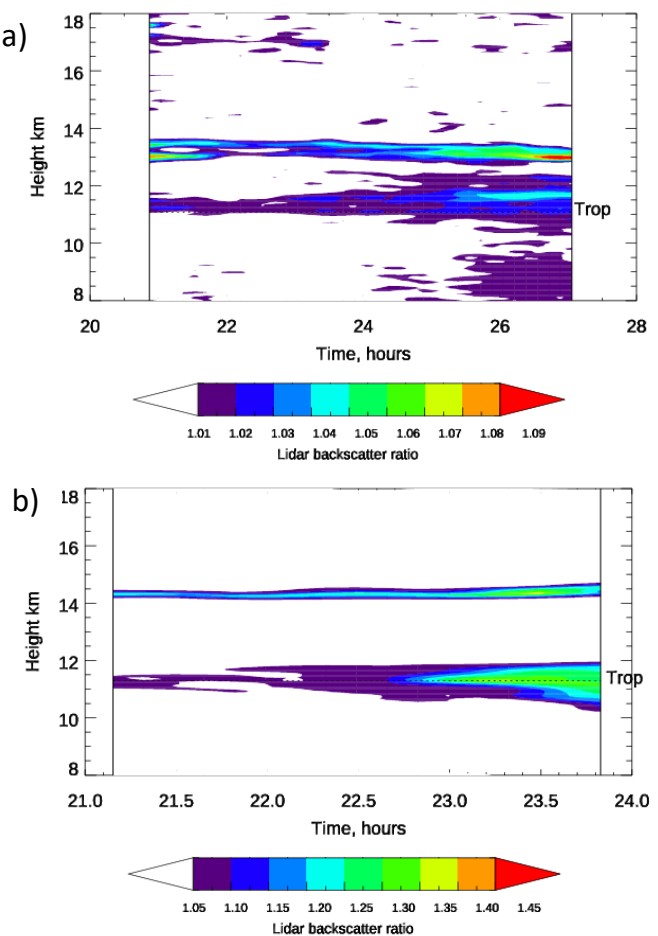

**Figure 3.** Lidar backscatter ratio at 355 nm measured at Capel Dewi during the nights of a) 1 July and b) 3 July 2019. Dotted black line denotes the tropopause height from the radiosonde at Valentia Observatory, Ireland (51.93°N, 10.25°W). Note the different colour scale on the two panels.

Consistent with the simulations of de Leeuw et al. (2020) and Kloss et al. (2020), the first unambiguous measurement of the volcanic aerosol cloud was on the night of 13-14 July (figure 5a), when a prominent layer of aerosol with peak backscatter ratio 1.4 was detected between 12.5 and 15 km. The figure shows the whole-night average (2130 - 0330 UTC); individual

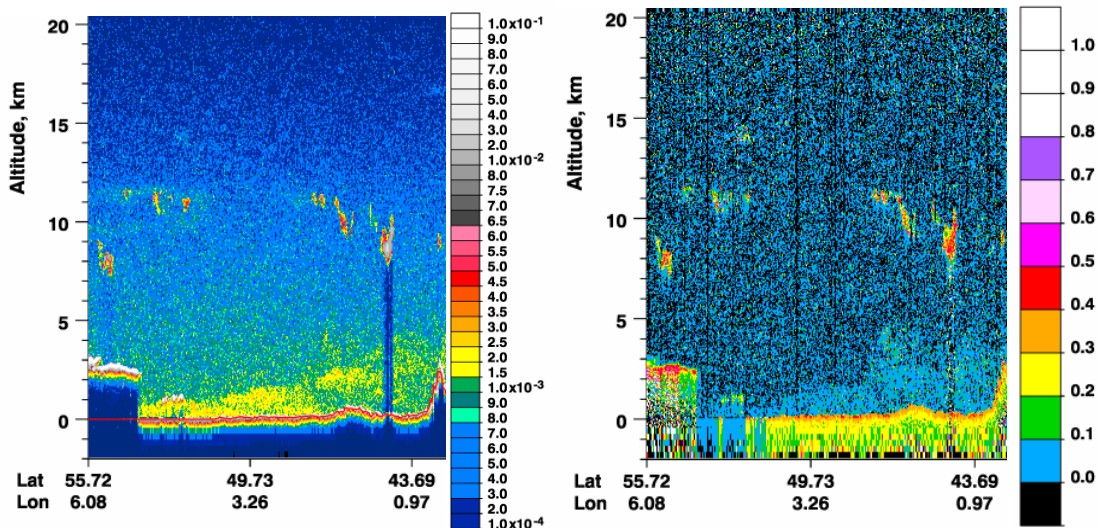

**Figure 4.** CALIOP lidar measurements along an orbit over central England between 0214 and 0217 UTC on 4 July 2019. Left panel: Total attenuated backscatter at 532 nm, in $sr^{-1}$ $km^{-1}$; right panel: depolarisation ratio. Image courtesy of NASA, https://www-calipso.larc.nasa.gov/products/lidar/browse_images/std_v4_showdate.php?browse_date=2019-07-04, accessed 31 Jan 2021.

profiles showed multiple thin layers 200-300 m thick lasting about an hour (corresponding to around 60 km in length with the wind speed of 18 $ms^{-1}$ measured by the sonde at 14 km). A CALIOP orbit passing to the east of the UK on this night measured stratospheric aerosol up to 15 km (figure 6), with very little depolarisation north of 47°N. This indicates that the aerosol consisted of spherical sulphuric acid droplets by this time, with little or no ash.

140    Measurements after 13 July are consistent with the continued presence of spherical sulphuric acid aerosol over the lidar site. The actual profiles were variable during the first 2-3 months after the eruption, with multiple layers in the height range 12–20 km. A notable example is shown in figure 5b, taken between 20:37 and 21:47 UTC on 25 August. This has a very prominent layer at 21 km - reminiscent of that seen by Chouza et al. (2020) at 26 km over Hawaii on 24 September, which they tracked back to the Kamchatka region. Although CALIOP profiles around 25-26 August showed numerous thin (< 1 km) aerosol

145  layers below 19 km, they showed nothing above 20 km in the vicinity of the UK. The wind speed at 21 km according to the Herstmonceux radiosonde at 00 UTC on the 26th was 5 $ms^{-1}$, so the layer in figure 5b could have been as small as 24 km in horizontal extent, perhaps explaining why CALIOP did not observe it. We note that Kloss et al. (2020) report OMPS satellite measurements reaching up to 22 km in the month after the eruption, which is consistent with this observation.

For consistency in the analysis from 13 July onwards, a lidar ratio of 40 sr was adopted, except for two nights (7 and 13

150  Sept) when a value of 50 sr was needed to return the backscatter ratio to 1 in the troposphere, where the Raman measurements indicated no aerosol. For non-depolarising aerosol, this is the actual value of lidar ratio, which falls within the range of 30 - 60 sr reported by Mattis et al. (2010).

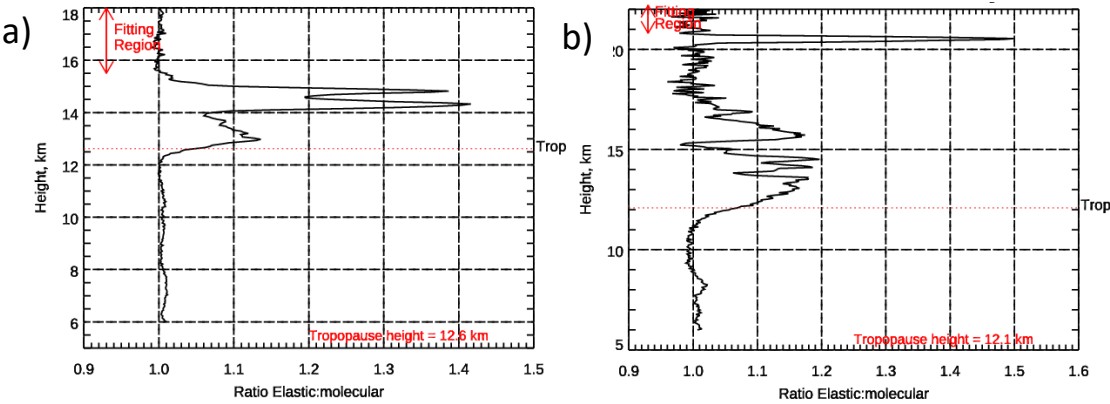

**Figure 5.** Average backscatter ratios at 355 nm for a) 13-14 July 2019 (21:31 - 03:22 UTC) and b) 25 August 2019 (20:37 - 21:47 UTC). The tropopause height as derived from the radiosonde is shown by the dotted red line.

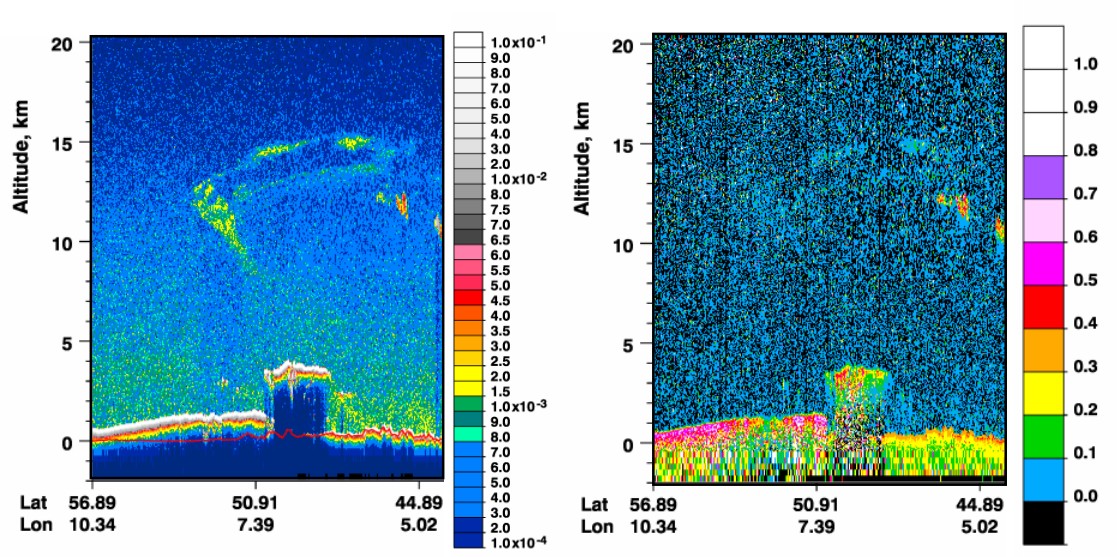

**Figure 6.** CALIPSO lidar measurements along an orbit to the east of the UK between 0210 and 0213 UTC on 14 July 2019. Left panel: Total attenuated backscatter at 532 nm, in $sr^{-1}$ $km^{-1}$; right panel: depolarisation ratio. Image courtesy of NASA, https://www-calipso.larc.nasa.gov/products/lidar/browse_images/std_v4_showdate.php?browse_date=2019-07-14, accessed 1 Aug 2020.

Figure 7a shows the evolution of the aerosol optical depth (AOD) above 12 km between 27 June 2019 and 30 May 2020 as measured by the Capel Dewi lidar. The optical depths reached around 0.05 by mid-August 2019, declining slowly for the remainder of the autumn, which is consistent with the results of Grebennikov et al. (2020) at 532 nm. An analysis of stratospheric AOD (sAOD) from the SAGE-III/ISS and TROPOMI satellites was presented by Kloss et al. (2020). They show

that the average sAOD from 55°N to 70°N measured by SAGE-III reached a maximum of 0.045 at 449 nm, falling to 0.030 at 1020 nm with an overall Angstrom exponent of 1.2. Corresponding sAOD values of 0.03 and 0.02 were measured for the latitude belt 40°N to 55°N. TROPOMI sAODs at 675 nm were about 10% lower than SAGE-III sAODs at 676 nm. (The corresponding maximum sAOD for Ulawun at 449 nm was 0.01 between 20°N and 20°S). As the Capel Dewi measurements are at 355 nm, a slightly higher peak sAOD of 0.050 is consistent with these other estimates.

Also shown on the figure are calculations of the aerosol optical depth above 12 km from the Raman channel, for nights clear enough to collect sufficient counts. In this analysis, an aerosol-free lidar profile was calculated from the radiosonde data and fitted to the Raman channel above the aerosol layer; the optical depth could then be derived from the ratio of the two profiles at 12 km. Even with long nights of data (and in the case of 4-6 February 2020, two nights combined data), the precision error bars on these estimates are large - but they are consistent with the estimates from the elastic channel, justifying the choice of lidar ratio. In contrast to the Raman estimates, where the precision error dominates, errors on the optical depth estimates from the elastic channel are dominated by the systematic uncertainty in the lidar ratio, since the precision errors are very small (and not plotted on the figure). For this reason, many lidar groups prefer to present their results as integrated backscatter (e.g. Trickl et al. (2013); Zuev et al. (2017); Grebennikov et al. (2020)), which does not depend directly on lidar ratio. Here we present the results as optical depth for comparison with the Raman measurements and because it is a more generally useful quantity. The uncertainty may be estimated from the range of lidar ratios used in our analysis of the volcanic aerosol cases – 40 – 50 sr. This implies an uncertainty of 20-25% in the optical depth measurements. Other than the two points marked on the figure, where the use of 50 sr best fitted the data (i.e. returned the profile to the molecular profile in the upper troposphere, where independent measurements from the Raman channel showed no aerosol), the use of 40 sr produced consistent analyses for the rest of the data set, suggesting that the particle size spectrum did not change substantially during the period of this study (Vaughan et al., 1994). For the period 27 June – 5 July 2019, when smoke aerosol was present in the upper troposphere and lower stratosphere, the value of 100 sr for lidar ratio is unrealistically high because the lidar is sensitive only to light polarised parallel to the laser. As it doesn't detect the perpendicular contribution to the backscatter, the measured aerosol backscatter ratios are too small. However, the method by which the lidar ratio is estimated, which involves returning the backscatter profile to the molecular value beneath the aerosol layer, compensates for the reduced backscatter and gives the correct optical depth values. These are therefore as accurate as in the later period ($\pm$ 20-25%).

The peak optical depth of around 0.05 was reached at the beginning of August 2019, declining to around 0.01 by the end of the year - an exponential decay time of around 3 months. The optical depth measurements reached a minimum of 0.008 on 4-5 February 2020, increasing slightly thereafter and reaching 0.014 on 20 May 2020.

Figure 7b shows how the maximum measured backscatter ratio for each night increased to a peak of 1.6 on 1 August 2019 before decreasing sharply to an average value of 1.045 in 2020. Figure 7c shows the height of the highest extent of the aerosol layer, the peak backscatter ratio and (for reference) the tropopause. The maximum height increased from around 15 km in mid-July to 20 km in September, remaining more or less constant thereafter at 20-21 km. The height of maximum backscatter ratio was more variable, with some outliers like that in figure 5b, but for the most part was around 15 km during 2019, with an apparent descent in 2020 to around 13 km.

## 5 Conclusions

The eruption of Raikoke on 22 June 2019 introduced a cloud of ash and sulphur dioxide into the lower stratosphere. For the first couple of weeks after the eruption the cloud remained in the general region between Kamchatka and Alaska, with the $SO_2$ oxidising to sulphuric acid in the form of spherical droplets (de Leeuw et al., 2020). Small patches of volcanic aerosol may have reached the UK in first few days of July, but were indistinguishable from the elevated aerosol background in the lower stratosphere at that time. The first unambiguous observation of volcanic aerosol at Capel Dewi, as suggested by the supplemental video of de Leeuw et al. (2020) showing the dispersion of the aerosol cloud, was therefore the night of 13-14 July 2019. CALIPSO profiles in the vicinity measured low depolarisation indicating that the cloud mostly consisted of spherical sulphuric acid droplets.

The measurements show that the aerosol optical depth between 12 and 21 km reached 0.05 at the beginning of August 2019, decaying to 0.01 by the end of 2019, and persisting up to May 2020 at around the same level. The maximum lidar backscatter ratio was 1.6 on 1 August 2019, with a sharp decrease reaching values < 1.1 from December onwards. It is likely that aerosol from the eruption of Ulawun in Papua New Guinea on 26 June 2019 mixed with the Raikoke aerosol over the months following the eruptions, so that the residual aerosol from August 2019 onward (Chouza et al., 2020) contained contributions from both sources.

*Data availability.* http://dx.doi.org/10.17632/6j67sfwkjx.1

*Author contributions.* D. Wareing performed the measurements, G. Vaughan and H. Ricketts performed the analysis and all authors contributed to writing the paper.

*Competing interests.* No competing interests

*Acknowledgements.* We thank Jim Haywood and Martin Osborne for fruitful discussions regarding the arrival of the volcanic aerosol over Europe, and to Vladislav Gerasimov and an anonymous reviewer for their thorough comments on the original manuscript. This work was supported by the Natural Environment Research Council through the National Centre for Atmospheric Science.

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

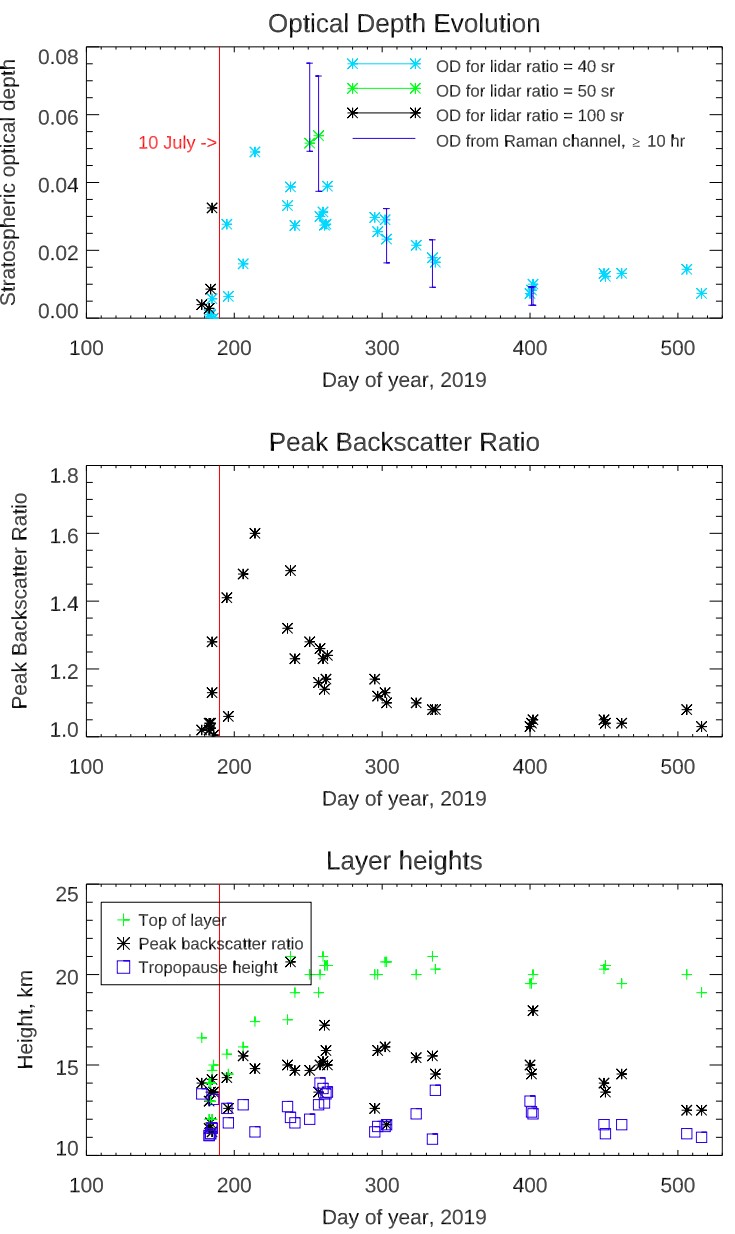

**Figure 7.** (a) Optical depth of the stratosphere at 355 nm between 27 June 2019 and 30 May 2020. Measurements after 10 July are considered to be volcanic aerosol, for which a lidar ratio of 40 or 50 sr was assumed (see text for discussion of errors on this plot). Earlier measurements are of smoke layers, where the lidar ratio has to be artificially increased to account for the depolarising particles. Also shown are estimates of optical depth from the Raman channel where more than 10 hours data was measured; the bars denote $\pm 1\sigma$ limits. (b) The corresponding peak backscatter ratio for each night (random error $\pm\ 0.02$). (c) The height of peak backscatter ratio (black asterisks), the top of the aerosol layer (green crosses) and tropopause height as calculated from a nearby radiosonde (blue squares).