# Peer review of "Measurement Report: Lidar measurements of stratospheric aerosol following the 2019 Raikoke and Ulawun volcanic eruptions"

_Atmospheric Chemistry and Physics, 2020_

## Referee Comment (RC1) · Vladislav Gerasimov (Referee) · 7 Nov 2020

**Manuscript ID:** acp-2020-982
**Type:** Measurement report
**Title:** Measurement Report: Lidar measurements of stratospheric aerosol following the Raikoke and Ulawun volcanic eruptions (Geraint Vaughan, David Wareing, and Hugo Ricketts)

**General comments**

The manuscript by G. Vaughan et al. reports on the stratospheric aerosol measurements performed after the 2019 Raikoke and Ulawun volcanic eruptions using the ground-based Raman lidar system at Capel Dewi Atmospheric Observatory, UK. The research results of the evolution of the aerosol optical depth above 12 km, backscatter ratio, and height of the highest extent of the aerosol layer between 27 June 2019 and 30 May 2020 are presented together with measurement data for the mentioned period. The authors also make an assumption concerning the origin of aerosol layers detected prior to the arrival of the Raikoke plume on 13-14 July 2019. The pyroconvection plumes ascended to the lower stratosphere over Canada are considered as a potential source of aerosol layers observed by the Capel Dewi lidar before 13-14 July.

The manuscript contains important data on lidar aerosol measurements referred to a certain measurement point (Capel Dewi) and made for a near one-year period (June 2019 – May 2020). Therefore, it will be of interest to the remote sensing and geoscience communities and can potentially be published as a Measurement report in ACP after some revision.

**Specific comments**

**1.** The manuscript contains several assumptions (even in the Abstract) that can be easily checked using forward or backward trajectory analysis. The first assumption is related with aerosol layers detected in the UTLS over Capel Dewi prior to the arrival of the Raikoke plume. For example:

**Citation 1** (Abstract, page 1, lines 6–7): "Small amounts of aerosol were measured prior to the arrival of the volcanic cloud, probably from pyroconvection over Canada."

**Citation 2** (Results, page 5, lines 92–93): "These are most likely due to pyroconvection, but it is not possible to rule out the arrival of volcanic aerosol over Europe at this time."

**Citation 3** contains the second assumption about aerosol layers detected in the stratosphere over Capel Dewi on 3 July 2019 and an assertion of unambiguous detection of volcanic ash on 13 July 2019 (Abstract, page 1, lines 7–8): "Volcanic ash may have first arrived as a thin layer at 14 km late on 3 July, and was certainly detected from 13 July onwards, eventually extending up to 20.5 km." See also (Conclusions, page 8, line 141): "The first unambiguous observation of volcanic aerosol at Capel Dewi was therefore the night of 13-14 July."

I do not insist, but I invite the authors to perform a trajectory analysis that will help to check the assumptions and prove the assertion for at least these two measurement dates (3 and 13 July 2019) like it was done, for example, by Vaughan et al. (2018), Fromm et al. (2010), Gerasimov et al. (2019), and Zuev et al. (2019). The trajectory analysis results can be added to the manuscript as "Supplement."

Vaughan, G., Draude, A. P., Ricketts, H. M. A., Schultz, D. M., Adam, M., Sugier, J., and Wareing, D. P.: Transport of Canadian forest fire smoke over the UK as observed by lidar, Atmos. Chem. Phys., 18, 11375–11388, https://doi.org/10.5194/acp-18-11375-2018, 2018.

Fromm, M., Lindsey, D.T., Servranckx, R., Yue, G., Trickl, T., Sica, R., Doucet, P., and Godin-Beekmann, S.: The Untold Story of Pyrocumulonimbus, B. Am. Meteorol. Soc., 91, 1193–1209, https://doi.org/10.1175/2010BAMS3004.1, 2010.

Gerasimov, V.V., Zuev, V.V. & Savelieva, E.S. Traces of Canadian Pyrocumulonimbus Clouds in the Stratosphere over Tomsk in June-July, 1991. Atmos Ocean Opt 32, 316–323 (2019). https://doi.org/10.1134/S1024856019030096.

Zuev, V. V., Gerasimov, V. V., Nevzorov, A. V., and Savelieva, E. S.: Lidar observations of pyrocumulonimbus smoke plumes in the UTLS over Tomsk (Western Siberia, Russia) from 2000 to 2017, Atmos. Chem. Phys., 19, 3341–3356, https://doi.org/10.5194/acp-19-3341-2019, 2019.

**2.** Another assumption in the manuscript ("It is likely that aerosol from the eruption of Ulawun in Papua New Guinea on 26 June 2019 mixed with the Raikoke aerosol over the months following the eruptions, so that the residual aerosol in 2020 contained contributions from both sources." (Conclusions, page 8, lines 144–146)) was proved by Chouza et al. (2020) (Fig. 7, page 6830). Therefore, the aerosol layers that were detected by the Capel Dewi lidar after August 2019 should contain volcanic plumes from both eruptions. The citation from Chouza et al. (2020): " Starting in July 2019, an enhancement of the aerosol load is clearly visible between the tropopause and 19 km a.s.l., with a small gap around 15∘ N. This gap corresponds to the division between the plume of the Ulawun (south) and Raikoke (north) eruptions. By August 2019, this gap is closed as both plumes mixed together, making them indistinguishable."

Chouza, F., Leblanc, T., Barnes, J., Brewer, M., Wang, P., and Koon, D.: Long-term (1999–2019) variability of stratospheric aerosol over Mauna Loa, Hawaii, as seen by two co-located lidars and satellite measurements, Atmos. Chem. Phys., 20, 6821–6839, https://doi.org/10.5194/acp-20-6821-2020, 2020.

**3.** I did not find in the manuscript any comparisons of the lidar measurements obtained by the authors with ones obtained by other lidar groups (excluding the results obtained in the middle of the Pacific (Chouza et al., 2020). Have similar aerosol measurements been made after the eruptions by lidar groups, for example, in Europe or somewhere else? I can recommend the authors to pay attention to the article about such lidar measurements that were made on the territory of Russia (Grebennikov et al., 2020).

Grebennikov, V.S., Zubachev, D.S., Korshunov, V.A. et al. Observations of Stratospheric Aerosol at Rosgidromet Lidar Stations after the Eruption of the Raikoke Volcano in June 2019. Atmos. Ocean Opt. 33, 519–523 (2020). https://doi.org/10.1134/S1024856020050097

**4.** The authors did not provide direct links to the CALIOP profiles and trajectories for the dates mentioned in the manuscript. This makes it difficult to read the article and to compare the CALIOP data with the authors' lidar measurements. I would recommend authors to provide a list of these links with the corresponding dates, for example, in the "Supplement."

An example of the direct link to the CALIOP data (not relevant to the authors' article):
https://www-calipso.larc.nasa.gov/products/lidar/browse_images/show_detail.php?s=production&v=V4-10&browse_date=2013-07-14&orbit_time=20-10-50&page=1&granule_name=CAL_LID_L1-Standard-V4-10.2013-07-14T20-10-50ZN.hdf

**Technical comments**

Find please my technical comments in the highlighted version of the manuscript below.

[revised manuscript text omitted]

---

## Referee Comment (RC2) · Anonymous Referee #2 · 6 Jan 2021

This manuscript presents an analysis of ground-based lidar measurements of the volcanic aerosol cloud from the June 2019 Raikoke eruption, made with a Raman lidar sited at the Capel Dewi atmospheric observatory, near Aberystwyth, Wales, U.K.

The study presents a limited set of data from the Capel Dewi lidar, with initial measurements on 1st and 3rd July 2019 showing clear layers of enhanced aerosol at 12km and 14km, and then layers detected from later lidar soundings at 14-15km on 13th/14th July and then at 20-21km on 25th August.

To interpret the layers detected from Capel Dewi, the authors also analyse spaceborne lidar measurements from the CALIPSO satellite, analysing profile-transects of

backscatter and depolarisation measured by the CALIOP lidar during orbits over the UK on 14th July, with also earlier the depolarisation within CALIOP lidar transects over the UK in earlier CALIPSO orbits.

The CALIOP analysis reveals the nature of the aerosol layers detected from the ground-based Raman lidar, with the 14th July measurements showing less than 1\% depolarisation, indicating spherical particles – then identified as aqueous sulphuric acid aerosol particles typical of ash-free portions of volcanic plumes. In contrast, the early July CALIOP profiles (not shown in the paper) show 10-20\% depolarisation in the lower 12km layer, the higher layer non-depolarising.

The authors present leave open the attribution to be either volcanic ash-sulphate mix from Raikoke or biomass smoke aerosol from the strong wildfire pyroconvection in Canada during June/July, which is also known to have also injected non-spherical particles into the stratosphere.

The authors then derive aerosol extinction from the aerosol backscatter assigning values for the extinction-to-backscatter-ratio (often referred to simply as "lidar ratio") of 40-50 steradians for the volcanic aerosol and 100 steradians for the biomass smoke.

The authors have chosen to present their results in the form of a measurement report, which then allows results and conclusions to be more limited in scope than would be required for a full ACP article:

https://www.atmospheric-chemistry-and-physics.net/about/manuscript_types.html

I am not sure whether the authors are aware, but there is currently a joint ACP/AMT/GMD Special Issue on "Satellite measurements, in-situ measurements and model simulations of the 2019 Raikoke eruption". Although the title of the Special Issue does not state that the non-satellite remote sensing observations such as those presented here are within scope, it would obviously make sense to include this manuscript within that special issue, and I am sure the ACP editors would welcome its inclusion.

In my comments, I am recommending the handling Topical Editor invite the authors to consider the manuscript can be re-aligned with the Raikoke Special Issue (assuming the ACP editors for the Raikoke SI agree, which I imagine they will) and request this manuscript and review to be included as an additional manuscript for the special issue.

In the spirit of that special issue, the reporting of the ground-based lidar measurements will represent a useful addition to the Special Issue. In particular, the link between the ground-based and space-borne lidar measurements represents an interesting analysis, potentially identifying an important difference between the later non-depolarising cloud and the earlier non-spherical particle layers detected in the early July ground-based lidar soundings.

The manuscript is mostly quite well written, and certainly in scope of the Measurement Report type ACP article the authors have chosen. I would recommend however that the authors consider potentially adding Figures showing the early July CALIOP layers, and also those to interpret the later August volcanic aerosol layer observed over Wales at the higher altitude of 20km.

The manuscript as currently presented also came across as a little disjointed, with the analysis of the progression of the cloud in the initial days after the eruption (with the HYSPLIT analysis in Figures 1 and 2) and seemed quite separate from the main topic of the article to analyse the layers observed from Aberystwyth ground-based lidar.

That said, I can see that the authors chose to include that to set the scene for the later analysis. Another of my comments refers the authors to another manuscript submitted to the Raikoke special issue (de Leeuw et al., 2020) which analyses simulations of the Raikoke cloud carried out with the NAME dispersion model.

It is interesting to contrast the initial progression of the cloud predicted by HYSPLIT with that seen in the NAME dispersion model, and there is a directly comparable Figure in the de Leeuw et al. (2020) manuscript for that 25th June date shown in Figure 1. It's clear that the NAME volcanic simulations (perhaps not surprisingly) predict much

greater dispersion of the cloud, with TROPOMI measurements of the SO2 confirming there was a lobe of the Raikoke SO2 cloud dispersed to the North West of the cyclonic recirculation, stretching already to the West of the original eruption location (153W).

Adding the extra CALIOP Figures, and some additional discussion re: differences between the HYSPLIT predicted dispersion and that seen in the TROPOMI observations (and represented in the NAME simulations from the de Leeuw et al. manuscript) would (in my opinion) enable the manuscript to be re-submitted as a full journal article to the Raikoke Special Issue.

However if the authors for some reason wish to keep the manuscript separate from the SI, or wish to re-allocate their submission to the SI retaining this Measurement Report type of manuscript that is also fine.

In my review below, I have listed a set of minor revisions that will improve the manuscript, and, once attended to, will make it publishable then in the SI as a Measurement Report.

These include being much clearer about the wavelength for the stratospheric AOD (sAOD) values derived from the ground-based lidar. Within my minor revisions below, I'm requesting to add a sentence referring to the OMPS and SAGE-III measuremnets of the Raikoke cloud shown in the Kloss et al. (2020) paper within the Raikoke special issue. The Figure there shows the sAOD from the 670nm SAGE-III channel to compare to that derived from OMPS (at 675nm), whereas the optical depth shown in Figure 6 is for the 355nm Raman channel. Since the authors may choose to retain their manuscript as this limited "Measurement Report" I am not requiring to compare directly to these other measurements, and the difference in wavelength makes that requiring to assume some Angstrom exponent for the wavelength translation, which itself is likely uncertain. The main comment here is to make sure the wavelength shown is communicated within the Figure captions so that the reader can always bear in mind the wavelength for the sAOD values being shown.

The authors also seem not be aware that recently the CALIOP team have developed a new stratospheric aerosol typing algorithm, which is described in Kim et al. (2018) and has a formalised categorisation into 4 types of stratospheric aerosol – polar stratospheric aerosol, volcanic ash, sulfate/other and smoke. This may help to formalise the choice of lidar ratio in the comparisons presented.

I've listed 20 specific revisions below, which although mostly minor in nature, do together comprise then major corrections. However, the authors can probably make those changes quite quickly, when I suggest they then re-submit to the Raikoke Special Issue, either retaining as a Measurement Report paper, or carrying out additional comparisons to other measurements such as TropOMI SO2 (Figure 1) and SAGE-III/OMPS sAOD measurements in Figure 6 – which would then elevate the status of the paper to have sufficient results to then be a "full ACP manuscript". The authors could also consider contacting Sergey Khaykin from Hautes Provence (lidar measurements shown in the SSiRC workshop's Raikoke page) and include some comparison between the timing of the plume detected over Wales and over Southern France.

Minor revisions ──────

1) Abstract line 1 – this 1st sentence should communicate more about the magnitude of the Raikoke eruption. The paper by Firstov et al. (2020) presents measurements of the infrasound wave signals from the eruptions from monitoring stations on the Kamchatka peninsula, which show the eruption had a volcanic explosivity index (VEI) of 4, i.e. it was a VEI4 eruption.

The Newhall and Self (1982) paper which presented the formalised basis for VEI estimation term VEI4 eruptions as "large magnitude eruptions" in contrast to eruptions at a VEI of 5 or larger being "very large eruptions". I recommend then the authors include to refer to the June 2019 Raikoke eruption as a "large magnitude eruption", consistent with the infrasound measurements from Firstov et al. (2020). The measurements show that there were 11 individual eruptions, with the 9th of these (from 23:00 on the 21st

June to 02:00 on the 22nd) the most explosive phase. I therefore suggest also to add "3-hour duration" to this initial sentence.

I suggest then to replace "erupted" with "began a 3-hour duration large magnitude explosive eruption", and change "On 22 June 2019" instead to "At 23 UTC on the 21st June 2019".

2) Page 1, lines 2-3 – Suggest to replace "has been used to measure" with "was deployed to measure", insert "vertical" before "extent" and either insert "enhancement to the" before "stratospheric aerosol layer" or replace with "volcanic aerosol cloud".

3) Page 1, line 5 – Replace "corrected for aerosol extinction" instead to "translated to aerosol extinction". It's not appropriate to refer to that as a correction – need to be clear that it's a conversion or translation of the measured backscatter into an aerosol extinction, based on an assumed lidar ratio (extinction to backscatter ratio), as is explained in the main part of the article. I would argue it is perhaps accurate even to refer to a "retrieval" of the aerosol extinction from the lidar's measurement of the backscatter. But suggest simply to replace "corrected for" to "translated to" or "converted to".

4) Page 1, line 5 – Change "comparison with aerosol-free profiles" instead to "subtracting the molecular backscatter profile (i.e. that from the gas phase)". The radiosondes enable to calculate the molecular backscatter, which then enables to isolate the aerosol backscatter signal from the total backscatter measured by the lidar.

5) Page 1, lines 6-8 – This sentence about the biomass burning smoke from the wildfires in Canada needs to be clearer the dates on which the Capel Dewi lidar observations are suggested to be wildfire smoke. The sentence after that states that the 14km layer in the 3rd July profile (Figure 3b) may have been volcanic ash. And the text in the main article states those layers in the 1st July and 3rd July lidar soundings (Figures 3a and 3b) could be caused either by the smoke aerosol from the Canadian wildfires or by volcanic ash from Raikoke. The text refers to CALIOP measurements from 2nd July and 4th July, but these are not shown in the manuscript, so it's not possible for

me as a reader to be able to make a judgment on which attribution is more likely. As I mentioned in my general comments above, I strongly recommend the authors add an extra Figure to show these CALIOP measurements – similar to Figure 5, but here perhaps it is possible to include these within Figure 3 as additional panels. This will really help the article to be able to understand the potential alternative attribution for these layers as either biomass smoke or volcanic ash (which would have within that air mass externally and internally-mixed volcanic sulphuric acid also.

Please clarify this in the Abstract text, being clear that the depolarisation from the CALIOP profile transects over the UK are being used to attribute the aerosol type measured from Capel Dewi. And please also re-draw Figure 3 adding the CALIOP transects from 2nd July and 4th July, and sharpen up the associated text discussing these early soundings.

6) Page 1, lines 8-9 – the language here is a little too informal here where it says "reached around 0.05 by early August" – the text needs to be precise in the Abstract here. Suggest to re-word the sentence with some additional context such as "A sustained period of clearly enhanced stratospheric Aerosol Optical Depth (sAOD) began in early August, with maximum sAOD at around 0.05 in mid-August, remaining above 0.02 until early November (around Julian day 310)."

7) Page 1, lines 9-10 – Replace "location of peak backscatter" with "altitude of peak backscatter" and insert "(between 14 and 18km)" after "considerably" to clearly summarise the variation seen in Figure 6.

8) Page 1, line 21 – Add a sentence here (after the text "previous 2 months.") referring to the OMPS and SAGE-III satellite measurements presented by Kloss et al. (2020), and re: the maximum values of the 675nm stratospheric AOD shown in the Figure 7 of that paper.

9) Page 2, line 25 – give some indication of the frequency at which the lidar measurements were made from Capel Dewi, and how many nights in each month of this period

the lidar soundings were made.

10) Page 2, line 41 – change "simulate an aerosol-free lidar profile" to "construct a molecular backscatter profile (for the gas phase)"

11) Page 2, line 42 – the authors refer to an "onion peeling algorithm" which I was not aware was being used in the retrieval of the backscatter ratio. My understanding is that once the molecular backscatter profile was determined, the aerosol backscatter was then simply determined from subtracting that molecular backscatter from the total backscatter measured. I am aware that one can account for the attenuation during the two-way transmission through the atmosphere, i.e. convert the "attenuated backscatter" to a "clean" aerosol backscatter (e.g. as explained in Young et al., 2015; and Antuna Marrero et al., 2020) which then accounts for the attenuation from aerosol and gas phase species in the two-way transmittance. Please add a sentence here to explain what is meant by the "onion-peeling algorithm.

12) Page 2, line 47 – The authors have stated "the choice of lidar ratio is to some extent arbitrary, since a wide range of values are given in the literature for aerosols of volcanic origin". But it is not correct that the choice is arbitrary, and also it does not follow from the remainder of that sentence. The choice of lidar ratio clearly affects the value of the aerosol extinction derived from the measured backscatter – and just because the values are variable or uncertain does not mean that the choice is necessarily arbitrary. I was very surprised at this sentence given that the lead author's 1994 GRL paper shows a very interesting Figure illustrating how the lidar ratio varies for a log-normal size distribution of sulphuric acid aerosol particles as one varies the geometric mean and geometric standard deviation for the mode (see Figure 6 of Vaughan et al., 1994). That Figure demonstrates how the size distribution alone can explain why the lidar ratio is variable between those values of 40 and 50. The presence of ash either externally mixed and/or internally mixed with the sulphuric acid aerosol will further affect the variation of the lidar ratio within a volcanic aerosol cloud. Please change that sentence from "arbitrary" to explaining "For volcanic aerosol clouds from very large magnitude

eruptions (e.g. 1991 Pinatubo) the lidar ratio for the cloud has been shown to be highly variable, for example due to variations in the size distribution of the sulphuric acid aerosol (e.g. Vaughan et al. 1994)". That sentence can follow after the sentence ending "volcanic origin.", replacing "is to some extent arbitrary" with "is known to be highly variable" and deleting "choice of" and changing "since a wide range..." to "with a wide range...". Then the sentence "For example, Mattis et al. (2010)..." can be revised to "For moderate magnitude eruptions, Mattis et al. (2010)..." so that it follows on from the sentence about very large magnitude eruptions.

13) Page 3, line 54 – The authors have stated "when the volcanic aerosol was bounded from above and below", and I think this must be referring to the method for calculating vertically integrated properties such as the stratospheric AOD. But it's far from clear what is meant there. Please re-word to clarify this.

14) Page 3, line 55 – the authors have written "an appropriate value of lidar ratio could be found by requiring that the backscatter return to the molecular profile below the layer." But that sentence does not make sense at all. Please revise the wording to explain what is meant.

15) Page 3, line 64 – the authors have stated "the ash and sulphur dioxide plume initially moved westward" – but I think the authors must mean "eastwards". Perhaps they are confusing this to "westerly" (i.e. from the west). Best to use eastwards though as that's less confusing to non-meteorological readers.

16) Page 3 line 66 – change "figure 1" to "Figure 1".

17) Page 3 line 67 – after "Kamchatka and Alaska." add a sentence explaining more about the HYSPLIT model. It needs to be communicated to the reader that this is not really a volcanic aerosol model – but just an isentropic air mass trajectory model. It should be made clear to the reader that the trajectories are indicative only, and not expected to be so accurate after several days. This is an interesting case that provides an opportunity to compare the volcanic plume dispersion shown in Figure 1 with equivalent simulations from the NAME dispersion model (used operationally for the London VAAC predictions) shown in Figure 6 of the manuscript by de Leeuw et al. (2020) submitted to the Raikoke special issue, currently in review for publication in ACP.

It is clear that the NAME dispersion model predicts quite different dispersion of the SO2, with a lobe of the volcanic cloud is transported to the west, which is not seen in the HYSPLIT predicted transport of the plume, the NAME predictions in good agreement with the TropOMI satellite measurements of SO2.

Please add a sentence acknowledging the simpler nature of the HYSPLIT predictions, and referring to this finding from the NAME dispersion model simulations that actual simulate the transport of the SO2 and ash, with vertical profile of emission reflecting an emission vertical profile from the VolRes community (see de Leeuw et al., 2020). And refer to the TropOMI observations showing the SO2 was dispersed also to the west of the Raikoke volcano as shown in Figure 6 of the de Leeuw et al. (2020) paper.

18) Page 4, lines 77-80 – The authors explain that in addition to the Raikoke volcanic aerosol cloud's dispersion, there is also biomass burning smoke from wildfires in Canada, which complicated attribution of the layers detected from Capel Dewi. The authors may not be aware however, that recently the CALIOP team have developed a new stratospheric aerosol typing algorithm, which is described in Kim et al. (2018) and has a formalised categorisation into 4 types of stratospheric aerosol – polar stratospheric aerosol, volcanic ash, sulfate/other and smoke. This may help to formalise the categorisation being explored here. Please add a sentence referring to this stratospheric aerosol type algorithm presented in Kim et al. (2018).

19) Page 5, line 97 – replace "lay" with "was detected".

20) Page 10, Figure 6 – Please redraw this Figure changing the symbols to be error bars showing the range in sAOD values derived for lidar ratio of 40 and 50. The two limit values will be relatively close together but in that sense represent a range based on assuming the layers are composed of volcanic aerosol. Then there should be two

values – one showing a range between the 40 and 50 lidar ratio derived sAOD (i.e. assuming volcanic) with the earlier points also having additional symbol for the 100 lidar ratio derived sAOD value (i.e. assuming biomass smoke).

References ———-

Firstov, P. P., Popov, O. E., Lobacheva, M. A., Budilov, D. I. and Akbashev, R. R. (2020) "Wave perturbations in the atmosphere accompanied the eruption fo the Raykoke volcano (Kuril Islands), 21-22 June 2019", Geosystems of Transition Zones, vol. 4, no. 1, pp. 1-11.

Kim, M.-H., Omar, A. H., Tackett, J. L., Vaughan, M. A., Winker, D. M., Trepte, C. R. et al. (2018) "The CALIPSO version 4 automated aerosol classification and lidar ratio selection algorithm" Atmos. Meas. Tech., vol. 11, pp. 6107–6135.

Kloss, C., Berthet, G., Sellitto, P., Ploeger, F., Taha, G., Tidiga, M. et al. (2020) "Stratospheric aerosol layer perturbations caused by the 2019 Raikoke and Ulawun eruptions and climate impact", Atmos. Chem. Phys. Discuss., https://doi.org/10.5194/acp-2020-701, in review.

de Leeuw, J., Schmidt, A., Witham, C. S., Theys, N., Taylor, I. A., Grainger, R. G. et al. (2020) "The 2019 Raikoke volcanic eruption: Part 1 Dispersion model simulations and satellite retrievals of volcanic sulfur dioxide", in review, Atmos. Chem. Phys., https://doi.org/10.5194/acp-2020-889.

Vaughan, G., Wareing, D. P., Jones, S. B., Thomas, L. and Larsen, N. "Lidar measurements of Mt. Pinatubo aerosols at Aberystwyth from August 1991 through March 1992", Geophys. Res. Lett., vol. 21, no. 13, pp. 1315-1318, 1994.

Young, S. A., Cope, M., Lee, S., Emmerson, K., Woodhouse, M. and Bellouin, N. (2015) "Simulation of cloud-aerosol lidar with orthogonal polarization (CALIOP) attenuated backscatter profiles using the Global Model of Aerosol Processes (GloMAP)", Extended Abstract to the 27th International Laser Radar Conference, New York, USA (July 2015). http://homepages.see.leeds.ac.uk/∼amtgwm/Young_etal_ExtAbs_IntLaserRadarConf_July2015.pdf

---

## Author Comment (AC1) · 10 Feb 2021

**Manuscript ID:** acp-2020-982
**Title:** Measurement Report: Lidar measurements of stratospheric aerosol following the Raikoke and Ulawun volcanic eruptions

Reply to Reviewer 1. Reviewer's comments in black, our comments in blue, new or amended text in red.

We thank Vladislav Gerasimov for his helpful comments on the paper.

**Comment 1:**
The manuscript contains several assumptions (even in the Abstract) that can be easily checked using forward or backward trajectory analysis. The first assumption is related with aerosol layers detected in the UTLS over Capel Dewi prior to the arrival of the Raikoke plume. For example:
**Citation 1** (Abstract, page 1, lines 6–7): "Small amounts of aerosol were measured prior to the arrival of the volcanic cloud, probably from pyroconvection over Canada."
**Citation 2** (Results, page 5, lines 92–93): "These are most likely due to pyroconvection, but it is not possible to rule out the arrival of volcanic aerosol over Europe at this time."
**Citation 3** contains the second assumption about aerosol layers detected in the stratosphere over Capel Dewi on 3 July 2019 and an assertion of unambiguous detection of volcanic ash on 13 July 2019 (Abstract, page 1, lines 7–8): "Volcanic ash may have first arrived as a thin layer at 14 km late on 3 July, and was certainly detected from 13 July onwards, eventually extending up to 20.5 km."
See also (Conclusions, page 8, line 141): "The first unambiguous observation of volcanic aerosol at Capel Dewi was therefore the night of 13-14 July."
I do not insist, but I invite the authors to perform a trajectory analysis that will help to check the assumptions and prove the assertion for at least these two measurement dates (3 and 13 July 2019) like it was done, for example, by Vaughan et al. (2018), Fromm et al. (2010), Gerasimov et al. (2019), and Zuev et al. (2019). The trajectory analysis results can be added to the manuscript as "Supplement."

We thank the reviewer for this suggestion. In fact, we did a trajectory analysis very similar to that published by Grebennikov et al (2020) in our initial analysis of this event. Unfortunately we found that the trajectories were very sensitive to initial conditions, and do not consider them accurate enough to include in this paper. Instead, we take advantage of the material in the supplement of de Leeuw et al (2020) to provide a more rigorous estimate of the spread of the volcanic cloud, with this text in section 3 (we also add a reference to Kloss et al (2020) for the spread of pyroconvection smoke in late June and early July):

The Hysplit model calculates air parcel trajectories based on 3-D advection by winds from an operational analysis model, and its predictions become increasingly sensitive to initial conditions as time goes on (e.g. Vaughan et al. (2018)). An alternative approach to simulating the spread of the aerosol cloud was presented by de Leeuw et al. (2020), using the UK Met Office's NAME dispersion model. This model is based on the global winds from the Met Office Unified Model analyses and includes chemical reactions for converting $SO_2$ to sulphate, as well as mixing through turbulence and subgridscale dynamics. Its simulations of $SO_2$ were found to agree well with the TROPOMI satellite for the three weeks after the eruption. de Leeuw et al. (2020) provide video files of model simulations as supplements to their paper, one of which shows the spread of volcanic aerosol across the Northern Hemisphere after the eruption. Up to the end of June the cloud was confined to North

America and eastern Asia. Between 1 and 4 July there are hints that small amounts of aerosol were reaching Europe, with a more prominent filament reaching Scotland by the 7th. The main aerosol cloud in this simulation reached the southern UK on 10 July. These conclusions are consistent with the CLAMS model simulations presented by Kloss et al. (2020) (their fig.5), suggesting that lidar observations over Europe might detect volcanic aerosol from 1 July onwards, and would definitely do so after the 10th. The analysis of OMPS satellite data by Kloss et al. (2020) showed small amounts of stratospheric aerosol over Europe 110 between 24 June and 6 July 2019 (their Fig. 3b), which they attribute using CLAMS modelling calculations to plumes from pyroconvection in Alberta.

**Comment 2**

Another assumption in the manuscript ("It is likely that aerosol from the eruption of Ulawun in Papua New Guinea on 26 June 2019 mixed with the Raikoke aerosol over the months following the eruptions, so that the residual aerosol in 2020 contained contributions from both sources." (Conclusions, page 8, lines 144–146)) was proved by Chouza et al. (2020) (Fig. 7, page 6830). Therefore, the aerosol layers that were detected by the Capel Dewi lidar after August 2019 should contain volcanic plumes from both eruptions.

Final sentence in conclusions now reads:
'It is likely that aerosol from the eruption of Ulawun in Papua New Guinea on 26 June 2019 mixed with the Raikoke aerosol over the months following the eruptions, so that the residual aerosol from August 2019 onward (Chouza et al., 2020) contained contributions from both sources.'

**Comment 3**

When we submitted the paper (in August) the Chouza et al paper was the only one we could find reporting lidar observations of Raikoke. We now include a paragraph on Grebennikov et al (2020) in the Introduction:
Lidar measurements of the volcanic aerosol cloud at 355 and 532 nm for four Russian stations were presented for the second half of 2019 by Grebennikov et al. (2020). These stations ranged in longitude from Obninsk at 36.6°E to Petropavlovsk-Kamchatsky at 158.65°E, and observed volcanic aerosol from late July onwards, reaching up to 18-20 km. A maximum integrated backscatter above 13 km of $> 10^{-3}$ was found in August, corresponding to aerosol optical depth of around 0.045.

**Comment 4**

The authors did not provide direct links to the CALIOP profiles and trajectories for the dates mentioned in the manuscript. This makes it difficult to read the article and to compare the CALIOP data with the authors' lidar measurements. I would recommend authors to provide a list of these links with the corresponding dates, for example, in the "Supplement."
A reference to the CALIPSO web page from which images may be browsed is given on the fifth line of section 3. It is an easy task to navigate from these to any desired image, if the time and date is known. An http reference has been added to the caption of fig 5 which shows a caliop image.

Technical comments
   a. Title. We have added 2019 to the title
   b. Abstract. Both reviewers asked for changes to the abstract, and we have mostly incorporated those of reviewer 2. For consistency we have removed the co-ordinates

of Capel Dewi from the Abstract and give all the coordinates in the main text, where the Ulawun eruption is also described. A sentence is included to describe the simulations of the spread of the aerosol cloud by de Leeuw etal (2020) and Kloss et al (2020). We also removed the 'probably' for the pyroconvection.

c.  Mann and Vernier reference replaced by Crafford and Venzke, 2019).

d.  Reference to Science article changed

e.  'Specificate please the month': following sentence added at the end of the Introduction: All the measurements were taken during the hours of darkness when there was no cloud cover over the site; in all there were 34 nights' measurements between 27 June 2019 and 30 May 2020.

f.  Westward corrected – now reads eastward (sect 3 l.1)

g.  We have not provided the direct link to the CALIOP image as explained above

h.  L.108 knots changed to $ms^{-1}$

i.  L. 114 – 2019 added and figure caption made consistent with text (thanks to the reviewer for spotting that mistake!)

j.  Pyroconvection over Canada: sentence removed from conclusions and replaced by: Small patches of volcanic aerosol may have reached the UK in first few days of July, but were indistinguishable from the elevated aerosol background in the lower stratosphere at that time.

k.  Trajectories: the reviewer has far greater faith in the accuracy of trajectory calculations than we do, so we have taken the published model simulations as our guide for the spread of the aerosol.

---

## Author Comment (AC2) · 10 Feb 2021

**Manuscript ID:** acp-2020-982
**Title:** Measurement Report: Lidar measurements of stratospheric aerosol following the Raikoke and Ulawun volcanic eruptions

Reply to Reviewer 2. Reviewer's comments in black, our comments in blue, new or amended text in red.

We thank the reviewer for their helpful comments on the paper.

Firstly, the reviewer suggests that this paper could be included in the joint ACP/AMT/GMD Special Issue on "Satellite measurements, in-situ measurements and model simulations of the 2019 Raikoke eruption". We were indeed not aware of this Special Issue when the paper was submitted to ACPD, and would like to accept the reviewer's suggestion that the paper be considered for that SI.

Secondly, the reviewer suggests adding some material to the paper and re-submitting it as a standard paper rather than a measurement report. We thank the reviewer for drawing our attention to de Leeuw et al and Kloss et al's papers in the SI which has allowed us to strengthen our discussion of the spread of the aerosol cloud considerably. In particular there is a supplement in de Leeuw et al (2020) which provides a NAME model simulation of the spread that improves considerably on what could be achieved by individual trajectory calculations. Nevertheless, we will keep this paper as a measurement report because we consider the measurements to be the scientifically useful part of this work.

It is interesting to contrast the initial progression of the cloud predicted by HYSPLIT with that seen in the NAME dispersion model, and there is a directly comparable Figure in the de Leeuw et al. (2020) manuscript for that 25th June date shown in Figure 1. It's clear that the NAME volcanic simulations (perhaps not surprisingly) predict much greater dispersion of the cloud, with TROPOMI measurements of the SO2 confirming there was a lobe of the Raikoke SO2 cloud dispersed to the North West of the cyclonic recirculation, stretching already to the West of the original eruption location (153W).

Adding the extra CALIOP Figures, and some additional discussion re: differences between the HYSPLIT predicted dispersion and that seen in the TROPOMI observations (and represented in the NAME simulations from the de Leeuw et al. manuscript) would (in my opinion) enable the manuscript to be re-submitted as a full journal article to the Raikoke Special Issue. However if the authors for some reason wish to keep the manuscript separate from the SI, or wish to re-allocate their submission to the SI retaining this Measurement Report type of manuscript that is also fine. In my review below, I have listed a set of minor revisions that will improve the manuscript, and, once attended to, will make it publishable then in the SI as a Measurement Report.

These include being much clearer about the wavelength for the stratospheric AOD (sAOD) values derived from the ground-based lidar. Within my minor revisions below, I'm requesting to add a sentence referring to the OMPS and SAGE-III measuremnets of the Raikoke cloud shown in the Kloss et al. (2020) paper within the Raikoke special issue. The Figure there shows the sAOD from the 670nm SAGE-III channel to compare to that derived from OMPS (at 675nm), whereas the optical depth shown in Figure 6 is for the 355nm Raman channel. Since the authors may choose to retain their manuscript as this limited "Measurement Report" I am not requiring to compare directly to these

other measurements, and the difference in wavelength makes that requiring to assume some Angstrom exponent for the wavelength translation, which itself is likely uncertain. The main comment here is to make sure the wavelength shown is communicated within the Figure captions so that the reader can always bear in mind the wavelength for the sAOD values being shown.

Reference to Kloss et al (2020) included multiple times in the revised paper, notably lines 138 and 146-151. Wavelength noted in captions of figures 3, 4 and 6.

The authors also seem not be aware that recently the CALIOP team have developed a new stratospheric aerosol typing algorithm, which is described in Kim et al. (2018) and has a formalised categorisation into 4 types of stratospheric aerosol – polar stratospheric aerosol, volcanic ash, sulfate/other and smoke. This may help to formalise the choice of lidar ratio in the comparisons presented.

We prefer to use the self-consistency of our own data to derive the lidar ratio independently, as CALIOP does not measure at 355 nm.

I've listed 20 specific revisions below, which although mostly minor in nature, do together comprise then major corrections. However, the authors can probably make those changes quite quickly, when I suggest they then re-submit to the Raikoke Special Issue, either retaining as a Measurement Report paper, or carrying out additional comparisons to other measurements such as TropOMI SO2 (Figure 1) and SAGEIII/ OMPS sAOD measurements in Figure 6 – which would then elevate the status of the paper to have sufficient results to then be a "full ACP manuscript". The authors could also consider contacting Sergey Khaykin from Hautes Provence (lidar measurements shown in the SSiRC workshop's Raikoke page) and include some comparison between the timing of the plume detected over Wales and over Southern France.

We will keep the paper as a measurement report.

Minor revisions ──────────
1) Abstract line 1 – this 1st sentence should communicate more about the magnitude of the Raikoke eruption. The paper by Firstov et al. (2020) presents measurements of the infrasound wave signals from the eruptions from monitoring stations on the Kamchatka peninsula, which show the eruption had a volcanic explosivity index (VEI) of 4, i.e. it was a VEI4 eruption.

The Newhall and Self (1982) paper which presented the formalised basis for VEI estimation term VEI4 eruptions as "large magnitude eruptions" in contrast to eruptions at a VEI of 5 or larger being "very large eruptions". I recommend then the authors include to refer to the June 2019 Raikoke eruption as a "large magnitude eruption", consistent with the infrasound measurements from Firstov et al. (2020). The measurements show that there were 11 individual eruptions, with the 9th of these (from 23:00 on the 21$^{st}$ June to 02:00 on the 22nd) the most explosive phase. I therefore suggest also to add "3-hour duration" to this initial sentence.

I suggest then to replace "erupted" with "began a 3-hour duration large magnitude explosive eruption", and change "On 22 June 2019" instead to "At 23 UTC on the 21st June 2019".

Having examined the information about the eruption more carefully, it seems that the explosive phase lasted more than 3 hours. We have included the following text in the Abstract:

At 18 UTC on 21 June 2019 the Raikoke volcano in the Kuril islands began a large magnitude explosive eruption, sending a plume of ash and sulphur dioxide into the stratosphere

And in the first line of the Introduction:

From 1805 UTC on 21 June 2019 to 0540 UTC on 22 June the Raikoke volcano in the Kuril Islands (48.29°N, 153.25°E) erupted, sending plumes of ash and sulphur dioxide into the stratosphere (Crafford and Venzke, 2019). With an estimated 1.5±0.2 Tg of SO2 (de Leeuw et al., 2020), it was one of the largest injections of volcanic aerosol into the stratosphere since20the Pinatubo eruption in 1991

2) Page 1, lines 2-3 – Suggest to replace "has been used to measure" with "was deployed to measure", insert "vertical" before "extent" and either insert "enhancement to the" before "stratospheric aerosol layer" or replace with "volcanic aerosol cloud".

Changed to: was deployed to measure the vertical extent and optical depth of the volcanic aerosol cloud following the eruption

3) Page 1, line 5 – Replace "corrected for aerosol extinction" instead to "translated to aerosol extinction". It's not appropriate to refer to that as a correction – need to be clear that it's a conversion or translation of the measured backscatter into an aerosol extinction, based on an assumed lidar ratio (extinction to backscatter ratio), as is explained in the main part of the article. I would argue it is perhaps accurate even to refer to a "retrieval" of the aerosol extinction from the lidar's measurement of the backscatter. But suggest simply to replace "corrected for" to "translated to" or "converted to".

Now reads 'retrievals of backscatter ratio profiles from the raw backscatter measurements required aerosol-free profiles derived from nearby radiosondes and allowance for aerosol extinction using a lidar ratio of 40-50 sr'.

4) Page 1, line 5 – Change "comparison with aerosol-free profiles" instead to "subtracting the molecular backscatter profile (i.e. that from the gas phase)". The radiosondes enable to calculate the molecular backscatter, which then enables to isolate the aerosol backscatter signal from the total backscatter measured by the lidar.

Done, see 3)

5) Page 1, lines 6-8 – This sentence about the biomass burning smoke from the wildfires in Canada needs to be clearer the dates on which the Capel Dewi lidar observations are suggested to be wildfire smoke.

Dates added

The sentence after that states that the 14km layer in the 3rd July profile (Figure 3b) may have been volcanic ash. And the text in the main article states those layers in the 1st July and 3rd July lidar soundings (Figures3a and 3b) could be caused either by the smoke aerosol from the Canadian wildfires or by volcanic ash from Raikoke.

We have included in our discussion of the NAME run in the supplement to de Leeuw et al (2020) more details about the dispersion of the cloud. Part 2 of that paper present an analysis of the spread of the Alberta fires to Europe.

The text refers to CALIOP measurements from 2nd July and 4th July, but these are not shown in the manuscript, so it's not possible for me as a reader to be able to make a judgment on which attribution is more likely. As I mentioned in my general comments above, I strongly recommend the authors add an extra Figure to show these CALIOP measurements – similar to Figure 5, but here perhaps it is possible to include these within Figure 3 as additional

panels. This will really help the article to be able to understand the potential alternative attribution for these layers as either biomass smoke or volcanic ash (which would have within that air mass externally and internally-mixed volcanic sulphuric acid also. Please clarify this in the Abstract text, being clear that the depolarisation from the CALIOP profile transects over the UK are being used to attribute the aerosol type measured from Capel Dewi. And please also re-draw Figure 3 adding the CALIOP transects from 2nd July and 4th July, and sharpen up the associated text discussing these early soundings.

Extra figure (fig 4) added showing CALIPSO data from early on 4 July 2019.

6) Page 1, lines 8-9 – the language here is a little too informal here where it says "reached around 0.05 by early August" – the text needs to be precise in the Abstract here. Suggest to re-word the sentence with some additional context such as "A sustained period of clearly enhanced stratospheric Aerosol Optical Depth (sAOD) began in early August, with maximum sAOD at around 0.05 in mid-August, remaining above 0.02 until early November (around Julian day 310)."

Now reads: A sustained period of clearly enhanced stratospheric Aerosol Optical Depths began in early August, with maximum value (at 355 nm) around 0.05 in mid-August and remaining above 0.02 until early November. Thereafter, optical depths decayed to around 0.01 by the end of 2019 and remained around that level until May 2020

7) Page 1, lines 9-10 – Replace "location of peak backscatter" with "altitude of peak backscatter" and insert "(between 14 and 18km)" after "considerably" to clearly summarise the variation seen in Figure 6.

Done

8) Page 1, line 21 – Add a sentence here (after the text "previous 2 months.") referring to the OMPS and SAGE-III satellite measurements presented by Kloss et al. (2020), and re: the maximum values of the 675nm stratospheric AOD shown in the Figure 7 of that paper.

We find the results in Kloss et al somewhat confusing, since figs 7 and 8 are not consistent with each other (In fig. 7b the maximum sAOD at 676 nm is 0.027 while in fig 8a it is around 0.037 for the highest latitude band). Line 473 of that paper states that 'For Raikoke, the whole average sAOD (plume plus background) reaches values as large as 0.045 (at 449 nm) to 0.030 (at 1020 nm), at 55-70°N and 0.030 to 0.020, at 40-55°N'. We quote this result in the Introduction in our revised manuscript:

The ash however was quickly dispersed or sedimented from the stratosphere in the week following the eruption, leaving the cloud of $SO_2$ to oxidise to sulphuric acid and spread around the hemisphere in aerosol form. Maximum average stratospheric aerosol optical depths at 449 nm (derived from SAGE-III/ISS data) reached 0.045 north of 55°N and 0.030 between 40 and 55°N in the early months after the eruption (Kloss et al., 2020)

9) Page 2, line 25 – give some indication of the frequency at which the lidar measurements were made from Capel Dewi, and how many nights in each month of this period the lidar soundings were made.

Done, following sentence added at end of Introduction:

All the measurements were taken during the hours of darkness when there was no cloud cover over the site; in all there were 34 nights' measurements between 27 June 2019 and 30 May 2020

10) Page 2, line 41 – change "simulate an aerosol-free lidar profile" to "construct a molecular backscatter profile (for the gas phase)"
changed to construct a molecular backscatter profile

11) Page 2, line 42 – the authors refer to an "onion peeling algorithm" which I was not aware was being used in the retrieval of the backscatter ratio. My understanding is that once the molecular backscatter profile was determined, the aerosol backscatter was then simply determined from subtracting that molecular backscatter from the total backscatter measured. I am aware that one can account for the attenuation during the two-way transmission through the atmosphere, i.e. convert the "attenuated backscatter" to a "clean" aerosol backscatter (e.g. as explained in Young et al., 2015; and Antuna Marrero et al., 2020) which then accounts for the attenuation from aerosol and gas phase species in the two-way transmittance. Please add a sentence here to explain what is meant by the "onion-peeling algorithm.
The reviewer's understanding is incorrect. To obtain the aerosol backscatter profile, allowance must be made for the attenuation of the laser beam by the aerosol itself. Often a Klett algorithm is used for this purpose, but we have used an equivalent method. The following text is now added:
An onion-peeling retrieval with prescribed lidar ratio (ratio of aerosol extinction tobackscatter coefficient) was then used to derive the lidar backscatter ratio down to the upper troposphere, as used in Thomas et al. (1987) and Vaughan et al. (1994). This algorithm sequentially removed the attenuation due to the aerosol, layer by layer, beginning from the aerosol-free fitting region and ending at 6 km altitude.

12) Page 2, line 47 – The authors have stated "the choice of lidar ratio is to some extent arbitrary, since a wide range of values are given in the literature for aerosols of volcanic origin". But it is not correct that the choice is arbitrary, and also it does not follow from the remainder of that sentence. The choice of lidar ratio clearly affects the value of the aerosol extinction derived from the measured backscatter – and just because the values are variable or uncertain does not mean that the choice is necessarily arbitrary. I was very surprised at this sentence given that the lead author's 1994 GRL paper shows a very interesting Figure illustrating how the lidar ratio varies for a log-normal size distribution of sulphuric acid aerosol particles as one varies the geometric mean and geometric standard deviation for the mode (see Figure 6 of Vaughan et al., 1994). That Figure demonstrates how the size distribution alone can explain why the lidar ratio is variable between those values of 40 and 50. The presence of ash either externally mixed and/or internally mixed with the sulphuric acid aerosol will further affect the variation of the lidar ratio within a volcanic aerosol cloud. Please change that sentence from "arbitrary" to explaining "For volcanic aerosol clouds from very large magnitude eruptions (e.g. 1991 Pinatubo) the lidar ratio for the cloud has been shown to be highly variable, for example due to variations in the size distribution of the sulphuric acid aerosol (e.g. Vaughan et al. 1994)". That sentence can follow after the sentence ending "volcanic origin.", replacing "is to some extent arbitrary" with "is known to be highly variable" and deleting "choice of" and changing "since a wide range..." to "with a wide range...". Then the sentence "For example, Mattis et al. (2010)..." can be revised to "For moderate magnitude eruptions, Mattis et al. (2010)..." so that it follows on from the sentence about very large magnitude eruptions.
The reviewer is engaging in semantics here. We have to assume a lidar ratio in order to perform the retrievals, and it is the choice of that ratio which is arbitrary, because of the large range of possible values. That is why we present results from the Raman channel analysis

which directly measures the AOD, to show that our estimates are consistent. Our previous work explored the variation of lidar ratio with aerosol properties (subject to the fairly restrictive assumptions of pure sulphate aerosol and a log normal size distribution) but we don't know the particle sizes in this case, or their composition. Nevertheless we have made the changes requested.

13) Page 3, line 54 – The authors have stated "when the volcanic aerosol was bounded from above and below", and I think this must be referring to the method for calculating vertically integrated properties such as the stratospheric AOD. But it's far from clear what is meant there. Please re-word to clarify this.
Changed to 'formed a distinct layer'

14) Page 3, line 55 – the authors have written "an appropriate value of lidar ratio could be found by requiring that the backscatter return to the molecular profile below the layer." But that sentence does not make sense at all. Please revise the wording to explain what is meant.
It doesn't make sense to the reviewer because they don't understand how a lidar retrieval works. Hopefully our answer to 11 above will clarify the method.

15) Page 3, line 64 – the authors have stated "the ash and sulphur dioxide plume initially moved westward" – but I think the authors must mean "eastwards". Perhaps they are confusing this to "westerly" (i.e. from the west). Best to use eastwards though as that's less confusing to non-meteorological readers.
Indeed we confused eastward and westerly. Sorry, now corrected.

16) Page 3 line 66 – change "figure 1" to "Figure 1".
Done

17) Page 3 line 67 – after "Kamchatka and Alaska." add a sentence explaining more about the HYSPLIT model. It needs to be communicated to the reader that this is not really a volcanic aerosol model – but just an isentropic air mass trajectory model. It should be made clear to the reader that the trajectories are indicative only, and not expected to be so accurate after several days. This is an interesting case that provides an opportunity to compare the volcanic plume dispersion shown in Figure 1 with equivalent simulations from the NAME dispersion model (used operationally for the London VAAC predictions) shown in Figure 6 of the manuscript by de Leeuw et al. (2020) submitted to the Raikoke special issue, currently in review for publication in ACP. It is clear that the NAME dispersion model predicts quite different dispersion of the SO2, with a lobe of the volcanic cloud is transported to the west, which is not seen in the HYSPLIT predicted transport of the plume, the NAME predictions in good agreement with the TropOMI satellite measurements of SO2.
Please add a sentence acknowledging the simpler nature of the HYSPLIT predictions, and referring to this finding from the NAME dispersion model simulations that actual simulate the transport of the SO2 and ash, with vertical profile of emission reflecting an emission vertical profile from the VolRes community (see de Leeuw et al., 2020). And refer to the TropOMI observations showing the SO2 was dispersed also to the west of the Raikoke volcano as shown in Figure 6 of the de Leeuw et al. (2020) paper.
We thank the reviewer for drawing our attention to de Leeuw et al (2020). The following paragraph now included in section 3:

The Hysplit model calculates air parcel trajectories based on 3-D advection by winds from an operational analysis model, and its predictions become increasingly sensitive to initial conditions as time goes on (e.g. Vaughan et al. (2018)). An alternative approach to simulating the spread of the aerosol cloud was presented by de Leeuw et al. (2020), using the UK Met Office's NAME dispersion model. This model is based on the global winds from the Met Office Unified Model analyses and includes chemical reactions for converting $SO_2$ to sulphate, as well as mixing through turbulence and subgridscale dynamics. Its simulations of $SO_2$ were found to agree well with the TROPOMI satellite for the three weeks after the eruption. de Leeuw et al. (2020) provide video files of model simulations as supplements to their paper, one of which shows the spread of volcanic aerosol across the Northern Hemisphere after the eruption. Up to the end of June the cloud was confined to North America and eastern Asia. Between 1 and 4 July there are hints that small amounts of aerosol was reaching Europe, with a more prominent filament reaching Scotland by the 7th. The main aerosol cloud in this simulation reached the southern UK on 10 July. These conclusions are consistent with the CLAMS model simulations presented by Kloss et al. (2020) (their fig.5), suggesting that lidar observations over Europe might detect volcanic aerosol from 1 July onwards, and would definitely do so after the 10th .

18) Page 4, lines 77-80 – The authors explain that in addition to the Raikoke volcanic aerosol cloud's dispersion, there is also biomass burning smoke from wildfires in Canada, which complicated attribution of the layers detected from Capel Dewi. The authors may not be aware however, that recently the CALIOP team have developed a new stratospheric aerosol typing algorithm, which is described in Kim et al. (2018) and has a formalised categorisation into 4 types of stratospheric aerosol – polar stratospheric aerosol, volcanic ash, sulfate/other and smoke. This may help to formalise the categorisation being explored here. Please add a sentence referring to this stratospheric aerosol type algorithm presented in Kim et al. (2018). We thank the reviewer for pointing this out and now include reference to Kim et al when discussing Fig. 3. However, the aerosol type in the CALIPSO plots is inconclusive in this case.

19) Page 5, line 97 – replace "lay" with "was detected". done

20) Page 10, Figure 6 – Please redraw this Figure changing the symbols to be error bars showing the range in sAOD values derived for lidar ratio of 40 and 50. The two limit values will be relatively close together but in that sense represent a range based on assuming the layers are composed of volcanic aerosol. Then there should be two values – one showing a range between the 40 and 50 lidar ratio derived sAOD (i.e. assuming volcanic) with the earlier points also having additional symbol for the 100 lidar ratio derived sAOD value (i.e. assuming biomass smoke). We do not agree with this suggestion, because it implies that the 'true' lidar ratio is $45 \pm 5$ sr, which is misleading – whereas the error bars on the Raman estimates are true precision standard errors. Furthermore, plotting the individual values as error bars also implies (by convention) that the errors in the different points are independent, whereas the uncertainty in lidar ratio is a systematic error. Nevertheless we agree that plotting measurements without error bars requires more explanation than we had provided and have added the following text: The uncertainty may be estimated from the range of lidar ratios used in our analysis of the volcanic aerosol cases – 40 – 50 sr. This implies an uncertainty of 20-25% in the optical depth measurements. Other than the two points marked on the figure, where the use of 50 sr

best fitted the data (i.e. returned the profile to the molecular profile in the upper troposphere, where independent measurements from the Raman channel showed no aerosol), the use of 40 sr produced consistent analyses for the rest of the data set, suggesting that the particle size spectrum did not change substantially during the period of this study (Vaughan et al 1994). For the period 27 June – 5 July 2019, when smoke aerosol was present in the upper troposphere and lower stratosphere, the value value of 100 sr for lidar ratio is unrealistically high because the lidar is sensitive only to light polarised parallel to the laser. As it doesn't detect the perpendicular contribution to the backscatter, the measured aerosol backscatter ratios are too small. However, the method by which the lidar ratio is estimated, which involves returning the backscatter profile to the molecular value beneath the aerosol layer, compensates for the reduced backscatter and gives the correct optical depth values. These are therefore as accurate as in the later period ($\pm$ 20-25%).

And added this to the figure caption:

see text for discussion of errors on this plot.

---

## Author Response (AR2)

**Manuscript ID:** acp-2020-982
**Title:** Measurement Report: Lidar measurements of stratospheric aerosol following the Raikoke and Ulawun volcanic eruptions

Author's response to Editor's comments

1. Reference for Hysplit changed.
2. Lines 51-58 now read: 'Raw data was collected with a time resolution of 10 minutes on most nights, and the files combined to whole-night averages for further analysis. Filters were applied during averaging to remove files affected by low cloud to guard against signal-induced noise problems. Analysis proceeded by converting the elastic signal profiles to lidar backscatter ratios - the ratio of the total backscatter profile to that which would be returned by a pure molecular atmosphere. The optimum way to accomplish this is to use data from the N2 Raman channel, as in Vaughan et al. (2018), as this automatically allows for attenuation of the signals by the aerosol layer. However, the faint signals on the N2 Raman channel in the lower stratosphere meant that long runs of data had to be combined to accumulate enough signal for analysis. This was only possible for a few nights during the period under consideration here.'